# Effects of Rowing on Rheological Properties of Blood

**DOI:** 10.3390/ijerph20065159

**Published:** 2023-03-15

**Authors:** Mateusz Mardyła, Aneta Teległów, Bartłomiej Ptaszek, Małgorzata Jekiełek, Grzegorz Mańko, Jakub Marchewka

**Affiliations:** 1Department of Physiology and Biochemistry, Institute of Biomedical Sciences, University of Physical Education in Krakow, 31-571 Krakow, Poland; 2Department of Health Promotion, Institute of Basic Sciences, University of Physical Education in Krakow, 31-571 Krakow, Poland; 3Institute of Applied Sciences, University of Physical Education in Krakow, 31-571 Krakow, Poland; 4Department of Physiotherapy, Faculty of Health Sciences, Jagiellonian University Collegium Medicum, 31-008 Krakow, Poland; 5Department of Biomechanics and Kinesiology, Institute of Physiotherapy, Jagiellonian University Collegium Medicum, 31-126 Krakow, Poland; 6ORNR “Krzeszowice”, Rehabilitation Center, Daszyńskiego 1, 32-065 Krzeszowice, Poland; 7Department of Rehabilitation in Traumatology, Institute of Clinical Rehabilitation, University of Physical Education in Krakow, 31-571 Krakow, Poland

**Keywords:** erythrocyte, rowing, exercise, blood rheology

## Abstract

The aim of this study was to analyze the selected hematological and rheological indices in female rowers during the competitive season. The study included 10 female rowers (aged 21.2 ± 2.6) and the control group consisted of 10 woman of corresponding age (non-athletes). The examination of athletes took place two times: at the beginning of the season during high endurance low intensity training period in January (baseline) and at the end of the competitive season in October (after). Blood samples taken from all woman were analyzed for hematological and rheological parameters. The training period of rowers during the 10 months resulted in decrease in red blood cell count and RBC deformability, in contrast to an improvement in some rheological functions such a decrease in fibrinogen concentration, plasma viscosity and aggregation index. The training program practice in rowing modulated some hematological and rheological indices. Some of them positively influenced the cardiovascular system and reduced potential risks connected with hard training and dehydration, but others may have followed from overtraining or not enough relaxation time between training units.

## 1. Introduction

Based on actual knowledge, there is no doubt that the role of physical activity on the human body is non-substantiable and many works discuss benefits in the prevention of many current diseases [1,2]. Nowadays, much attention concentrates on the protective impact of regular activity on cardiovascular disease as well as cardiovascular adaptation [3]. Due to physical effort, there is increased demand for oxygen, glucose and nutrients. This, in the long run, improves metabolic status connected with the transformation of lipids and carbohydrates [4]. Additional benefits from regular exercise alternate with improvements in blood rheology properties in some clinical situations [5,6]. As well as in medical and sports training, the objective is to improve muscle strength, increase the aerobic capacity and metabolic profile [7]; however, the goals of an athlete’s training program are to increase their sporting abilities. Thus, the efficiency of muscle, circulatory and hematopoiesis systems play an important role in determining the possibilities to achieve high results in sport competitions. Nonetheless, studies present that strenuous exercise might result in cardiovascular incidents in professional athletes [8,9,10].

One of these dangers could be some periodic negative changes related to hemorheology. Due to physical exercise, the blood drop hemoconcentration phenomenon can occur resulting in a loss of extracellular water [11,12]. Red blood cells are almost the half amount of the whole blood volume, and they affect the circulatory fluid dynamics. Blood, regarding its mechanical relevance, is a non-Newtonian liquid, which means that an increase in shear rate (depends on vessel radius and speed of blood flow) causes a decrease in viscosity [11]. It has direct meaning regarding the speed of blood flow. The hemorheology effects of training are seen to be dependent on the time spent training, and relate mainly to short, medium and long term exercise [4].

Erythrocytes (RBC) are characterized by their deformability and aggregation properties. RBC have the ability to shape and reshape erythrocytes and, thanks to this, they can pass the capillaries with a smaller diameter than theirs [13]. Single physical efforts lead to increased heart rate, cardiac output and accordingly this accelerates blood flow to the respiratory and exercising muscles. During enhanced flow in the cardiovascular system, erythrocytes are subject to many forces, mainly in the arteries. Even an increase in hematocrit to 60% will increase blood viscosity, but erythrocytes carry off the capillary system. Red blood cells’ deformability depends on membrane flexibility, cell geometry and internal cytoplasmic viscosity [14]. One of the deformability measurement procedures abuses a laser diffraction method as a fast and simple way to perform an elongation index of red cells [15]. In normal conditions, it is expected to decrease erythrocyte aggregation with reduced deformability, especially in low fibrinogen concentrations [11]. Erythrocytes could bind with each other and form aggregates, called rouleaux formation, and also 3D networks. This process is reversible. When higher shear forces occur in blood vessels, negative surface charge erythrocytes could disaggregate on single cells. RBC aggregation, which is strictly linked with blood flow especially in smaller venous compartments, could be the main agent of vessel resistance. Higher erythrocyte aggregation could increase vascular flow resistance and indirectly induce stronger work of the heart muscle, leading to the development of hypertension [16]. This can lead to cardiovascular consequences as well as some thrombotic diseases as a result of dehydration, increased blood coagulate, high physical work or contraceptive medications [17,18]. Nonetheless, regular physical exercise leads to an improvement in nitric oxide (NO) production by the endothelium and release from red blood cells [18]. As a result, this can develop protective mechanisms such as the regulation of arteriolar vasodilation.

To our knowledge, the training effects of the whole rowing season on rheological blood properties was not studied before. Only the work of Teległów et al. [19] described the effects of intermittent hypoxic training among professional male rowers on performance, rheological properties of blood and metabolic activity of erythrocytes. It is worth emphasizing that hematological research is usually conducted on male athletes because of the many factors that affect making accurate conclusions among women.

Thus, the present study aimed to investigate red blood cell deformability, aggregation, plasma viscosity and concentration of fibrinogen in professional female rowers, in different periods of the season.

## 2. Materials and Methods

### 2.1. Participants

The study involved 11 women who trained as professional rowers at the University Sport Krakow Club. The women were aged 18–25 years (mean = 21). In the main season, when the most championships were held, they trained for a total of about 8–10 training units per week, which consisted of main training on-water, gym exercise, ergometer and running. The sportswomen trained all year, except for 3 weeks in August and a few days at the end of December. In the period of the most important sports events (from May to September), they started about every other weekend. To assess the potential changes in blood morphology and rheology, the authors decided to take the samples two times (in January and October) from each competitor. Blood was collected from the ulnar vein, on an empty stomach in the morning hours. In addition, the control group consisted of 10 healthy, non-training women. The research was approved by the Bioethics Committee at the Regional Medical Chamber in Cracow. The participants were also informed about the aim of the research, and they approved their participation. The samples were taken at the Laboratory of Blood Physiology at the University of Physical Education in Cracow. A qualified nurse collected blood into 2 types of Vacuette test tubes: with potassium EDTA and with clot activator for serum. Blood samples were tested for morphological and rheological properties. Blood was used for examination one hour after collecting samples.

The measurements of the basic hematological indicators were performed on the HORIBA ABX Micros 60 (USA).

WBC [109/l]—White Blood Cells;RBC [1012/l]—Red Blood Cell;HGB [g/dl]—Hemoglobin;HCT [%]—Hematocrit;PLT [109/l]—Platelet Count;MCH [pg]—Mean Corpuscular Hemoglobin;MCV [fl]—Mean Corpuscular Volume;MCHC [g/dl]—Mean Corpuscular Hemoglobin Concentration.

### 2.2. Measurement of Erythrocyte Elongation and Aggregation

To study deformability and aggregation indicators, blood samples were analyzed in the LORRCA device (Laser-Optical Rotational Cell Analyzer (R&R Mechatronics, Hoorn, The Netherlands). For the determination of erythrocyte elongation, they have been put into 5 mL of standardized viscous solution Polyvinylpyrrolidone (PVP), M = 360,000, osmolality ~300 mOsm/kg. The temperature in Lorca was configured at 37 °C. Then, a sample was injected into the Lorca measuring system and was subject automatically to increasing shear stresses (from 0.3 Pa to 60.30 Pa). The system analyzed the diffraction of light on the blood cells and calculated the elongation index from the special formula:$$EI=\frac{a-b}{a+b}$$

*a*—is length of the red blood cell;

*b*—is width of the red blood cell.

To study aggregation, the same device (LORRCA) was used. A total of 1 mL of blood was previously subject to oxygenation for 10 min. Then, blood was injected into a rotating cylinder in LORCA. The computer was adjusted at 37 °C and it started moving the cylinder with a shear rate of >400 s^−1^. After 10 s, the cylinder stopped and the aggregation of red blood cells came out. Changes in the intensity of light diffraction and time when they occurred are presented on the syllectogram. The following parameters determining erythrocyte aggregation kinetics were assessed:$$AI=\frac{A}{A+B}\times 100\%$$

*AI* [%]—aggregation index;

*A*—area above syllectogram curve;

*B*—area below syllectogram curve.

Also analyzed were

AMP [au]—total extent of aggregation;

T½ [s]—half time kinetics of aggregation.

### 2.3. Determination of Fibrinogen Concentration

To indicate plasma fibrin concentration, the Chrom 7-coagulometer (Bio-Ksel, Grudziądz, Poland) was used. To 50 µL of plasma, we added 100 µL of Bio-Ksel PT reagent containing thromboplastin with calcium chloride. This resulted in plasma coagulation and clot formation. After conversion of fibrinogen to fibrin, the light detector analyzed the concentration of fibrinogen.

### 2.4. Blood Plasma Viscosity (BPV)

To examine plasma viscosity, we used the Myrenne Roetgen viscometer (D-52159, Myrenne GMBH, Roetgen, Germany). Before measuring the plasma viscosity, calibration was performed using two standard solutions Myrenne NP1 and NP2 for the lower and upper ranges of standard measurements, accordingly, for 1.10 mPas and 1.90 mPas. Then, 0.5 mL of plasma was placed into the measurement capillary. The device detected the time at which plasma passed through the distance from barrier L3 to L4 spectrophotometer, with constant pressure and temperature (37 °C). The normal range for human plasma viscosity is 1.10–1.90 mPas.

### 2.5. Statistical Analysis

Continuous variables are presented as mean ± standard deviation (SD) or median and interquartile range (IQR), depending on the normality of distribution. The normality of distribution was tested using the Shapiro–Wilk test. For intergroup comparisons, we used ANOVA or, in the case of not meeting its assumptions, Kruskal–Wallis test followed by post hoc tests: Tukey or Dunn, respectively. Dunnett’s Multiple Comparison was additionally used for testing with respect to the control group. SS1/2 and EImax were calculated by fitting SS versus EI to equation, representing Lineweaver–Burke model, using a non-linear, curve-fitting algorithm available in a commercial statistical package (Prism 7.02, GraphPad Software Inc., La Jolla, CA, USA). The methodology was described in detail by Baskurt et al. (2013). Calculations were performed using Statistica 12 (StatSoft^®^, Tulsa, OK, USA) software. Statistical significance was defined as *p* ≤ 0.053.

## 3. Results

In the study group, we found a significance decrease in red blood cell count (RBC) by 7.82%. Additionally, there was a higher erythrocyte level reported in the control group. Mean corpuscular volume (MCV) was higher in the athlete group compared to untrained individuals. Eight months of training resulted in an increase in average mass corpuscular hemoglobin (MCH) by 4.82%. The two groups did not differ significantly in the levels of hematocrit (Hct) and hemoglobin (Hgb). The results of the basic hematological measurements are listed in Table 1. Comparing rowers to the control group, we have found statistically significant changes: average RBC count decreased by 7.23% in rowers; average value of MCV (fl) increased by 6.19% in rowers at the first examination (baseline) versus controls; average value of MCV (fl) increased by 3.70 % in rowers at the second examination (after) versus controls; average mass of corpuscular hemoglobin MCH (fmol) increased by 4.82 % in rowers at the second examination (after) versus at first examination (baseline). In hemoglobin and fibrinogen concentration as well as WBC and PLT count, there were not any significant changes in rowers compared to the control group.

Blood plasma viscosity

The viscosity in the athlete group was meaningfully lower at both examinations compared to the control group (*p* < 0.05). Compared to the first examination, there was a notable decrease in blood plasma viscosity of 13.85 % in rowers (baseline) compared to the control group and of 17.70% in rowers (after) compared to the control group (see Table 2 below).

Erythrocyte deformability

The eight-month professional training was reflected by a significantly lower Elongation Index (EI) in the athlete group at a shear stress level between 0.30 Pa and 15.98 Pa; however, at higher levels (31.03 and 60.3 Pa), there were no differences observed. In a group of non-training women (control), there was a higher erythrocyte deformability noted at all shear stress levels compared to the athlete group (see Table 3 below).

Aggregation parameters

The degree of total aggregation (AMP) was found to be significantly higher in athletes compared to control groups. We did not observe any significant changes in terms of Aggregation Index (AI) and half time of total aggregation (t1/2) in athletes and controls.

## 4. Discussion

The main objective of this study was to examine the effects of the changes in the resting morphological and rheological properties of blood in response to a long-term training period. There are enormous data about the impact of different exercises on basic blood morphological parameters and the effects of different energetic character, duration and other factors existing in specific sport disciplines. In rowing, applying high loads can lead to a disturbance of the morphotic elements in the bloodstream. At the beginning of the season, in rowers, the main objective of the implemented training is to enhance general performance and therefore aerobic capacity. Therefore, this time serves to enhance aerobic capacity, among other things, by the increase in red blood cells coming into the circulation. The training process in rowing is divided into classical periods proposed by Matvejev, such as preparatory, starting and transitional [20]. The samples were collected at the beginning of the preparatory period (baseline), where a significant part of the training provided aerobic schemes with cross country skiing, gym exercise and rowing indoors (60–70% of time). The second collection took place after the last national competition, at a moment of significant decrease in intensity and volume. This is the time when in some athletes the overtraining symptoms can occur.

The total number of circulating red blood cells and total hemoglobin concentration are mostly responsible for oxygen transport to muscles via the cardiovascular system [12]. These factors are particularly important in professional athletes competing mainly in endurance sports such as cycling, cross-country skiing, long distance running, rowing and many others. However, RBCs rheological properties such as deformability and aggregation could change as a result of intensive physical training periods. Physical exercise affects the increase in demand and consumption of oxygen by the working tissues (mainly muscles and neurons). The best athletes should have high oxygen uptake to tolerate exhaustive efforts, without any risks for their health. RBC homeostasis is related to changes in hematopoiesis systems, including RBC production, RBC hemolysis, bone marrow activity and iron resources [21]. Basic hematological indicators such as hemoglobin, red blood cell count and hematocrit are strictly connected with the maximal oxygen uptake, which is one of the determinants of aerobic capacity [22]. In general, it is proven that endurance training results in a positive increase in red blood cell count, both in men and women [23]. Nonetheless, it is known that in women, the monthly menstruation cycle can disturb blood circulation, levels of erythrocytes or hemoglobin, and some rheological properties [24,25]. Additionally, hemoglobin concentration in women is strictly linked to the inhibitory impact of estrogens (estradiol) on bone marrow activity [26]. Although the examinations were placed in the intermenstrual cycle, it cannot be excluded for sure that the changes were related to monthly blood loss through menstruation, as well as amenorrhea or oligomenorrhea which is frequent in athletes. It is also worth underlining that regular endurance training seems to accelerate the selective elimination of rigid and old RBCs via the reticuloendothelial macrophage system [27]. The level of reduced hematological indices such as HGB, HCT and iron also could be influenced by the dietary intake, both with sport-related issues, such as the type of sport (endurance/power), and the amount of training [28,29]. The most recent data suggest that supplementation with iron preparations could fill the iron stores (serum ferritin), contributing to simultaneously increasing the energetic efficiency status in female rowers [30].

It is known that highly-trained athletes have changed parameters related to the erythrocytic system, such as higher mean corpuscular volume (MCV), higher hemoglobin concentration, higher 2,3 2,3-bisphosphoglycerate and adenosine triphosphate (ATP) in erythrocytes compared to non-trained subjects [31]. The distinct values enumerate red cell indices with a quicker turnover in hematopoietic systems. Under the regular physical training, there was an observed drop in mean corpuscular hemoglobin concentration (MCHC), which is strictly related to an increase in red blood cell elasticity.

Diminished erythrocyte deformability can occur as the action of reactive oxygen species (ROS) intensifies during hard physical exercise or during long exposure to hypoxic conditions [32,33]. Certain researchers have underlined the effects of the adaptation of trained subjects (humans and mice), and no instance of erythrocyte damage has been found as a result of oxidative stress [34,35]. Decreased levels of HGB and PLT, as well as PV, could indicate that there is more plasma (as an effect of training). Interesting findings were provided by the Dunch research team [36], who compared female and male rowers with blood donor candidates. The retrospective analysis performed by comparing the hemoglobin levels in healthy people and athletes confirmed the higher concentration in both male and female athletes than in donors. Hemoglobin concentration measured photometrically, in 10.4% of male rowers and 8.3% of female rowers, was above the recommended level for the competition [36]; whereas, only 3.9% male and 1.9% female blood donors exceeded these values. A hemoglobin concentration of 10.5 mM equals a hematocrit of 51% in males and 9.7 mM equals 47% in females. Boyadijev and Taralov [37] found significant differences in mean red blood cell count, with it being lower in highly trained athletes generally compared to untrained subjects. The same research evidence on lower hematological indices such as hemoglobin and hematocrit and higher mean corpuscular volume was found in female highly trained athletes compared to untrained people. In agreement with this, the authors came to a similar conclusion. In the study of Dellavalle et al. [29], considering iron status, the hematological variables including Hb, Hct and mean cell volume were close to those reported in our work. The research of Skarpańska-Stejnborn et al. [38] demonstrates that under the influence of intensive exercise in rowers, there are disturbances in iron resources as a result of increased escape through macrophages. Some explorers discovered that also in other groups of athletes, such as footballers, there was a depletion of iron, independent of the training system, which can occur with lower concentration of hemoglobin. In our research, we could not detect these findings because there was not a measured level of iron in the serum. In the literature, we can find different procedures of measurement deformability index. In the present work, we used a method based on a laser diffractometer through the Lorrca device. This could be the reason for little changes in our results in comparison to other researchers who used alternative methods. The results of this paper indicated that regular training performed by professional rowers led to significant alterations in blood plasma viscosity, connected likely with the shift in blood compounds and plasma volume. At both time points, in rowers, there were significant lower plasma viscosity levels compared to untrained individuals. Many research conclusions approved the thesis that under the impact of regular endurance training, there was an enhanced volume of plasma. The reason for that is an increase in albumin concentration, followed by shift from the extracellular space [39]. What is interesting is that four females were tested shortly after 3 weeks of mountain training, which could be another proof of an activity that increases plasma volume and decreases its viscosity. This reaction is strictly connected with the renin–angiotensin–aldosterone axis. Although in the discussed article the authors did not measure the whole plasma volume, it may have corresponded with plasma viscosity. The earlier works concerning this issue [40,41] conclude that plasma viscosity is lower in athletes than in untrained subjects, which depends mainly on fibrinogen concentration. Other measured indices such as plasma total protein, globulin and albumin were also lower in runners, but their impact was probably not strong enough to modulate plasma viscosity. Additionally, plasma renin activity was significantly lower in runners, which corresponds with a lower heart rate and adrenergic tone than was previously described by the other authors [39].

Athletes who train excessively could be exposed to overtraining syndromes (OTS). The main symptoms associated with hemorheology could be the feeling of heavy legs, occurring with concomitant modifications in some hematological parameters such as decrease in red blood cells or hemoglobin. A study by Varlet-Marie et al. showed that overtraining syndromes could be connected with hemorheology disturbances such as mild hyperaggregation and mild hyperviscosity [42]; however, in this study, only a higher amplitude of aggregation was noted in female rowers at the end of the season, without increasing plasma viscosity, suggesting that athletes did not experience OTS. It could be related to a loss of water, increase in hematocrit, increase in sodium and potassium ions in the serum, nitrate in urine and transaminase, as a result of hard training (catabolic intensification) [43]. In natural conditions, the increase in hematocrit correlates with decreased performance; however, it is not the rule in every condition.

In athletes, the better indicator of OTS experience is increased plasma viscosity. In our research, we noted a significant decrease in plasma viscosity in the experimental group. There was also a relationship during physical exercise between deformability of red blood cells and deficit of oxygen which is used in energetic processes. This leads to the formation of free radicals and may disturb the flexibility of the RBC membrane [4]. Despite antioxidant mechanisms (RBC have about 30 different enzymes), they are not free from oxidative damage. Additionally, the close neighborhood of leukocytes, activated through exercise, can lead to the structural alteration of erythrocytes [44].

We acknowledge some potential limitations in our study. The limitation of our study is that we did not examine hemorheology changes at different time points during all seasons in athletes. We think that a comparison with similarly practicing rowing men athletes would also provide valuable information on blood rheology and differences/convergence between genders. Moreover, we did not analyze performance indicators (such as cardio-respiratory gas ergospirometry), which could be helpful in the evaluation of training processes and their relationship to blood morphology.

## 5. Conclusions

In our results, we did not find any significant changes in the comprehensive overview of blood hemorheology, but we could see some single modifications such as in plasma viscosity, red cell volume and aggregation. In the examined group, ensuring consistency between the time of blood collection, age of athletes and training experience led to us obtaining stable results with marginal deviations between athletes.

## Figures and Tables

**Table 1 ijerph-20-05159-t001:** Mean values (±SD) of hematological parameters and fibrinogen concentration in the rowers at first and second examination compared to the control group.

Parameter	Rowers (Baseline)	Rowers (after)	*p* (Baseline vs. after)	Control Group	*p* (after vs. Control)
RBC [1012]/L	4.46 ± 0.40	4.23 ± 0.21	0.041 *	4.56 ± 0.28	0.012 *
HGB [g/dl]	13.31 ± 1.15	13.25 ± 0.61	0.690	13.53 ± 0.85	0.452
HCT [L/L]	39.39 ± 3.96	39.89 ± 1.78	0.703	40.65 ± 2.39	0.849
MCV [fl]	88.25 ± 3.10	86.18 ± 2.44	0.031 *	83.10 ± 3.60	0.002 *
MCH [pg]	29.85 ± 1.10	31.36 ± 0.92	0.007 *	29.65 ± 1.26	0.013 *
MCHC [mmol/L]	33.82 ± 0.90	33.24 ± 0.48	0.163	33.23 ± 0.56	0.952
WBC 10^9^/L	5.53 ± 1.55	5.75 ± 0.89	0.125	6.72 ± 1.42	0.083
PLT 10^9^/L	262.55 ± 88.17	246.55 ± 48.99	0.810	253.90 ± 30.06	0.999
Fibrinogen [g/dl]	4.6 ± 0.75	3.93 ± 0.76	0.041 *	4.59 ± 1.02	0.475

* *p* < 0.05.

**Table 2 ijerph-20-05159-t002:** Median (with interquartile ranges) of blood plasma viscosity (BPV) in rowers at first and second examination compared with control group.

Parameter	Rowers (Baseline)	Rowers (after)	*p* (Baseline vs. after)	Control Group	*p* (after vs. Control)
BPV [mPas]	1.12 (0.26)	1.07 (0.12)	0.188	1.30 (0.14)	0.036 *

* *p* < 0.05.

**Table 3 ijerph-20-05159-t003:** Mean values of Elongation Index (EI) depends on shear stress and aggregation parameters.

Parameter	Rowers (Baseline)	Rowers (after)	*p* (Baseline vs. after)	Control Group	*p* (after vs. Control)
EI 0.30 [Pa]	0.03 ± 0.02	0.05 ± 0.01	0.036 *	0.06 ± 0.01	0.001 *
EI 0.58 [Pa]	0.06 ± 0.01	0.07 ± 0.01	0.003 *	0.09 ± 0.01	0.001 *
EI 1.13 [Pa]	0.12 ± 0.01	0.13 ± 0.02	0.007 *	0.15 ± 0.01	0.001 *
EI 2.19 [Pa]	0.22 ± 0.02	0.21 ± 0.02	0.081	0.24 ± 0.01	0.002 *
EI 4.24 [Pa]	0.33 ± 0.03	0.30 ± 0.03	0.008 *	0.33 ± 0.02	0.026 *
EI 8.24 [Pa]	0.41 ± 0.03	0.37 ± 0.03	0.004 *	0.41 ± 0.03	0.034 *
EI 15.98 [Pa]	0.47 ± 0.03	0.44 ± 0.03	0.005 *	0.48 ± 0.03	0.009 *
EI 31.03 [Pa]	0.52 ± 0.03	0.51 ± 0.02	0.370	0.54 ± 0.02	0.031 *
EI 60.3 [Pa]	0.55 ± 0.03	0.55 ± 0.02	0.095	0.58 ± 0.02	0.051 *
AMP [au]	14.69 ± 3.64	15.43 ± 5.26	0.062	10.05 ± 2.38	0.019 *
AI [%]	57.58 ± 10.24	54.01 ± 6.64	0.385	58.04 ± 6.38	0.584
t1/2 [s]	2.95 ± 1.32	3.20 ± 1.00	0.884	2.63 ± 0.64	0.394

* *p* < 0.05.

## Data Availability

The datasets used and/or analyzed during the current study are available from the corresponding author on reasonable request.

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
