# Peer review of "Effects of Rowing on Rheological Properties of Blood"

_ijerph, 2023, doi:10.3390/ijerph20065159_

Round 1
Reviewer 1 Report
I would like to thank Mardyła et al. for the opportunity to review the manuscript. "Effects of rowing on rheological properties of blood". In this article, the authors set themselves the goal - to investigate red blood cell deformability, aggregation, plasma viscosity and concentration of fibrinogen in professional female rowers in different periods of season. Using modern equipment, the authors studied these parameters at the beginning of the training and competition season and at its end. As a result of the study, they obtained some scientific facts, showed that they did not find any significant changes in the comprehensive review of blood hemorehology in athletes. However, changes in parameters such as plasma viscosity, RBC volume, and aggregation have been identified. When reviewing the manuscript, I had questions and comments. 1. I have the feeling that there is no scientific novelty in this study. Indeed, out of 42 publications in the list of references, 2/3 are more than 10 years old, and 29% are older than the last century. This may indicate that the research topic is not relevant, since there are very few publications on it in recent years. When finalizing the article, the authors need to more clearly show what the novelty of their research is. For example, the same group of authors studies blood rheological parameters in different groups of athletes (1.2), in other cohorts (3.4). Are there any new data obtained in this work in comparison with the previously obtained results? There are other recent studies evaluating rheology in athletes and during exercise (5). 2. When analyzing the results, the authors actually compare three groups (athletes twice, control). In this case, it is not enough to evaluate the differences between groups in pairs, it is necessary to use the ANOVA or Kruskal-Wallice tests. 3. The authors did not state the limitations of the study, there is no clearly formulated conclusion of the study. 4. The Discussion section is written chaotically, it should be rewritten and start with the main results obtained by the authors and then compare them with the results of modern research without references to the work of past decades. References: 1. Teległów A, Marchewka J, Tota Ł, Mucha D, Ptaszek B, Makuch R, Mucha D. Changes in blood rheological properties and biochemical markers after participation in the XTERRA Poland triathlon competition. Sci Rep. 2022 Mar 1;12(1):3349. doi: 10.1038/s41598-022-07240-1. 2. Teległów A, Dąbrowski Z, Marchewka A, Tyka A, Krawczyk M, Głodzik J, Szyguła Z, Mleczko E, Bilski J, Tyka A, Tabarowski Z, Czepiel J, Filar-Mierzwa K. The influence of winter swimming on the rheological properties of blood. Clin Hemorheol Microcirc. 2014;57(2):119-27. doi: 10.3233/CH-141823. 3. Marchewka A, Filar-Mierzwa K, Dąbrowski Z, Teległó A. Effects of rhythmic exercise performed to music on the rheological properties of blood in women over 60 years of age. Clin Hemorheol Microcirc. 2015;60(4):363-73. doi: 10.3233/CH-131793. 4. Filar-Mierzwa K, Marchewka A, Dąbrowski Z, Bac A, Marchewka J. Effects of dance movement therapy on the rheological properties of blood in elderly women. Clin Hemorheol Microcirc. 2019;72(2):211-219. doi: 10.3233/CH-180470. 5. Nader E, Guillot N, Lavorel L, Hancco I, Fort R, Stauffer E, Renoux C, Joly P, Germain M, Connes P. Eryptosis and hemorheological responses to maximal exercise in athletes: Comparison between running and cycling. Scand J Med Sci Sports. 2018 May;28(5):1532-1540. doi: 10.1111/sms.13059. |
Author Response
Dear Reviewer,
Thank you for your revision. I'd like to comment about your questions.
Point 1. At first I'd like to note that most of published articles are focused on effects of exercise, training or participation in competiton on hemorheology properties and especially in male athletes. There is less literature which analyse this point at sportswoman. Here we compared pre season and after season indices of blood morphology and rheology and we have tried to find correlation between training loads and changes of these parameters. We remember that female profile of blood can change due to menstrual cycle and we consider that for planning blood collection. The novelty of this work I think is related with assay of plasma viscosity which is very rare in previous publication related to this subject.
Point 2. We recalculated data using ANOVA, or in case of not meeting its assumptions Kruskal-Wallis test followed by post-hoc tests (Tukey or Dunn respectively) and we obtained similar statistical results as presented in paper. Dunnett’s Multiple Comparison was additionally used for testing to control group. Therefore it is possible to use multiple t-tests or its non-parametric counterparts to asses changes between dependent and nondependent variables especially in small sample size and with double – sided p.
Point 3. The limitation of study is that we didn't examine hemorheology changes in different time points during all season in athletes. We think that compared with similarly practice rowing men athletes would also provide valuable information on blood rheology and diffrences/ convergence between gender.
Point 4. I agree that some part of discussion need to be rewritten .
Best regards,
Mateusz Mardyła
Reviewer 2 Report
The authors of the article presented interesting and statically valid results. The manuscript is well-organized and easy to read.
However, I have a few questions for the authors.
1. Based on the data in Table 1, the average cell volume in October (86.18 ± 2.44) was lower than in January (88.25 ± 3.10), while the hemoglobin content (MCH) increased over this period (31.36 ± 0.92 versus 29.85 ± 1.10). These changes should increase the MCHC value, but this is not observed. What do the authors think causes this?
2. I recommend that the authors add a paragraph to the Discussion section addressing the limitations of this study.
I recommend the manuscript for publication after appropriate changes and clarifications have been made.
Author Response
Dear Reviewer,
Thank you for Your revision. I would like to answer on your questions.
Point 1. We have discussed together about this values related to mean hemoglobin concentration in erythrocytes but unfortunately we cannot find any certain reason why this shift doesnt't occured. The possible reason could be related to changing of internal viscosity of cells affecting the result of MCHC.
Point 2.
The limitation of study is that we didn't examine hemorheology changes in different time points during all season in athletes. We think that compared with similarly practice rowing men athletes would also provide valuable information on blood rheology and diffrences/ convergence between gender.
Best regards,
Mateusz Mardyła
Round 2
Reviewer 1 Report
The authors answered questions and made some corrections to the manuscript. However, I still have questions that I would like responces.
1. The answer to the first question does not satisfy me. The introduction contains well-known information with references to old studies, which rather confirms the lack of relevance of the study, rather than vice versa. In response to the question, the authors tried to justify the relevance of the study, but this is not enough, it is required to adjust the Introduction section accordingly.
2. I do not agree with the author's answer to the second question. When comparing the three groups, it is required to use adequate statistical methods (ie, ANOVA or Kruskal-Wallis test). The use of the method of pairwise comparison of groups in such cases is incorrect.
4. The authors corrected the Discussion section somewhat, but they still use mostly old publications in the links. It should be recognized that the authors in a strange way confirm the relevance of their research. In my opinion, this section needs further correction.
Author Response
Dear Reviewer,
As you wish we revised our manuscript consider your cues, especially about statistical analysis.
Best regards
Mateusz Mardyla
